# Cytotoxicity Comparison of ^99m^Tc-Labeled Peptide Antagonist and Agonist Targeting the SSTR2 Receptor in AR42J Cells

**DOI:** 10.3390/molecules30081715

**Published:** 2025-04-11

**Authors:** Sahar Nosrati Shanjani, Monika Łyczko, Rafał Walczak, Przemysław Koźmiński, Emilia Majka, Jerzy Narbutt, Wioletta Wojdowska, Agnieszka Majkowska-Pilip, Aleksander Bilewicz

**Affiliations:** 1Institute of Nuclear Chemistry and Technology, Dorodna 16, 03-195 Warsaw, Poland; sahar.nosrati.shanjani@gmail.com (S.N.S.); m.lyczko@ichtj.waw.pl (M.Ł.); r.walczak@ichtj.waw.pl (R.W.); p.kozminski@ichtj.waw.pl (P.K.); e.amajka@ichtj.waw.pl (E.M.); j.narbut@ichtj.waw.pl (J.N.); 2National Centre for Nuclear Research, Sołtana 7/3, 05-400 Otwock, Poland; wioletta.wojdowska@polatom.pl

**Keywords:** radionuclide therapy, auger electron, ^99m^Tc radiopharmaceuticals

## Abstract

Auger electrons are low-energy, high-linear-energy-transfer particles that deposit their energy over nanometers distances. Their biological impact depends heavily on where the radionuclide is localized within the cell. To verify the hypothesis that the cell membrane may be a better molecular target than the cytoplasm in Auger electron therapy, we investigated whether the radiotoxicity of ^99m^Tc varied depending on its location in the cell. The behavior of peptide radiopharmaceuticals ^99m^Tc-TECANT-1 targeted the cell membrane was compared with ^99m^Tc-TEKTROTYD directed to the cytoplasm. Our findings confirmed that ^99m^Tc-TECANT-1 displayed greater binding to AR-42-J cells than ^99m^Tc-TEKTROTYD. Additionally, it was demonstrated that the receptor agonist ^99m^Tc-TEKTROTYD is localized in more than 90% of the cytoplasm, while ^99m^Tc-TECANT-1 is found in 60–80% of the cell membrane. When evaluating cell survival using the MTS assay, we observed that toxicity was significantly higher when ^99m^Tc was targeted to the membrane compared to the cytoplasm. This indicates that, for ^99m^Tc, as with ^161^Tb, the membrane is a more sensitive target for Auger electrons than the cytoplasm. Our results also suggest that receptor antagonists labelled with therapeutic doses of ^99m^Tc may be effective in treating certain cancers. However, further detailed studies, particularly dosimetric investigations, are necessary to validate these findings.

## 1. Introduction

Cancer is one of the significant factors affecting life expectancy. According to a 2019 report by the World Health Organization (WHO), cancer is the leading cause of death in individuals under the age of 70 in 112 out of 183 countries and ranks third or fourth in an additional 23 countries. The standard primary cancer therapeutic options include surgery, chemotherapy, and radiotherapy. In the 21st century, the use of radionuclides in the combined diagnosis and treatment of cancer is becoming increasingly common. Alpha and beta emitters are current choices for targeted radionuclide therapy. β^−^-emitters are suitable for larger and solid cancers because of the relatively long-range emissions. However, part of the released energy is deposited in the surrounding normal tissues relative to the tumor. α-particles have a much shorter range and greater linear energy transfer (LET), depositing more energy into smaller volumes, ideal for small tumors and metastasis, but with some limitations. The first limitation is the small number of radionuclides with suitable properties for nuclear medicine purposes and their availability. The second one is that most of the α-emitters suitable for therapy have a decay chain with multiple radioactive daughters, leading to unnecessary exposure of healthy tissues to the daughters if they are not well controlled.

Targeted Auger electron therapy can successfully replace α therapy. Radionuclides that emit Auger electrons (AE) are much cheaper and more readily available than α-emitters. Targeted Auger electron therapy is also more accurate and causes less damage to adjacent healthy cells. The linear energy transfer (LET) of Auger electrons ranges from 1 to 23 keV/µm [1]; therefore, Auger electrons are similar to α particles and produce highly damaging effects in cells. The energy of Auger electrons is deposited over nanometer distances, resulting in high LET that is potent for causing lethal damage in cancer cells by inducing double-stranded DNA breaks [2]. It is possible to inject approximately tenfold the radioactivity of Auger emitters compared to β^−^ particle emitters without toxic side effects, particularly affecting the bone marrow. Certainly, this is especially relevant for Auger electron emitters, which do not emit high-energy and intense gamma quanta, such as the recently proposed radionuclides ^103^Pd/^103m^Rh, ^58^Co, ^193m,195m^Pt or ^119^Sb. Therefore, it might be assumed that radiopharmaceuticals labeled with Auger emitters will gain widespread application in radionuclide therapy in the near future. This assumption is grounded on the significant cytotoxicity and therapeutic efficacy reported, as well as the availability of several low-energy electron-emitting radionuclides in non-carrier-added forms, characterized by variable physical half-lives and established chemical properties [2].

In therapy using AE, one of the key factors limiting the broader application of this method is the need to precisely deliver the radionuclide, which is the emitter of Auger electrons, to the interior of the cell nucleus. This requires the development of a radiopharmaceutical that will recognize cancer cells and also allow the isotope to be placed within the cell nucleus in the immediate vicinity of the DNA strand. However, recent studies postulated that since the cell and nucleus membranes have a critical function for cell survival, the effects of AE emitted by membrane-bound radiolabeled biomolecules also need to be considered [3]. One approach to targeting the cell membrane using AE emitters involves attaching a radioactive isotope to a molecule, such as an antibody, peptide, or small molecule, that specifically binds to cancer cell receptors. When the isotope decays, it emits Auger electrons that can damage cell membranes and associated proteins if the isotope is in close proximity. If the Auger electrons interact with the lipid bilayer or membrane proteins, they could cause localized damage that may disrupt membrane integrity. This could lead to the loss of membrane function and ion imbalances and lead to the death of cancer cells, which opens up new perspectives in the design of effective radiotherapeutic methods [3].

Neuroendocrine tumors (NETs) are a group of tumors that are characterized by the overexpression of somatostatin receptors (SSTR), particularly SSTR type 2. This overexpression allows for the use of molecular imaging and radionuclide therapy for NET tumors. Radiolabeled somatostatin analogs, such as TOC, TATE, and NOC, which are conjugated with the chelator DOTA, are commonly applied for this purpose. Mentioned conjugates act as SSTR agonists and are internalized into the cytoplasm through the receptor. Several reviews have detailed the use of these radiolabeled agents in the management of NETs [4]. Although ^68^Ga-labelled somatostatin analogues are the gold standard of NETs, in the PET technique ^99m^Tc-labelled TOC and TATE are also commonly used in the SPECT diagnosis of NETs.

On the other hand, LM3 (p-Cl-Phe- cyclo(D-Cys-Tyr-D-4-amino-Phe(carbamoyl)-Lys-Thr-Cys)D-Tyr- NH_2_) is an SSTR antagonist. The conjugate of DOTA with LM3 is often labeled with ^68^Ga for NETs diagnosis and ^177^Lu for therapeutic purposes [5]. As demonstrated with ^68^Ga-DOTA-LM3, LM3 radioconjugates demonstrate favorable biodistribution, high tumor uptake and retention, and also minimal safety concerns. In a study involving 40 patients, ^68^Ga-DOTA-LM3 showed greater diagnostic efficacy than the agonist ^68^Ga-DOTATATE [6]. The first human study in Zentralklin Bad Berka of the SSTR antagonist, ^177^Lu-DOTA-LM3, for radionuclide therapy has demonstrated significant efficacy in treating advanced metastatic NETs [7]. This new SSTR antagonist shows favorable biodistribution and higher tumor uptake compared to the SSTR agonist, ^177^Lu-DOTATOC, and has even resulted in complete remission for some patients.

The advantages of the LM3 antagonist became more evident when comparing the antagonist ^161^Tb-DOTA-LM3 with the agonist ^161^Tb-DOTATATE. Similar to ^177^Lu, ^161^Tb emits soft beta radiation and additionally releases 10.9 Auger electrons per decay [8,9]. The observed higher dose-response may stem from the significantly higher membrane-bound fraction of ^161^Tb-DOTA-LM3, along with the emission of Auger electrons from ^161^Tb, which causes additional damage to the cell membrane. Consequently, the SSTR2 antagonist ^161^Tb-DOTA-LM3 demonstrated greater therapeutic efficacy for labeling with ^161^Tb compared to the agonist ^161^Tb-DOTATATE, making ^161^Tb-DOTA-LM3 the preferred choice for clinical translation.

In this study, we explored the potential of using the widely applied imaging radionuclide ^99m^Tc as an Auger electron emitter for the treatment of NETs. We used the ^99m^Tc-labeled LM3 peptide, which acts as an antagonist to the SSTR receptor, targeting the cell membrane as the site of action.

While ^99m^Tc is commonly used for diagnostic imaging, particularly in Single Photon Emission Computed Tomography (SPECT), the potential for Auger electron emission could offer therapeutic benefits. According to Eckerman and Endo [10], the electrons emitted from the core shells of ^99m^Tc can be categorized as follows: Auger electrons, which have a yield of 4.4 per decay and internal conversion electrons of 1.1 per decay. So, on average, approximately 5.5 electrons are emitted from the atomic shells. ^99m^Tc is not an ideal AE source for targeted radiotherapy because it has a relatively low yield of emitted electrons and high γ emission. Despite this, the low price and wide availability of the ^99^Mo/^99m^Tc generator suggest that ^99m^Tc could be used in Auger electron therapy. However, the primary objective of our study is to confirm the hypothesis that in Auger electron therapy, the cell membrane serves as a more effective therapeutic target than the cytoplasm. Previous research on this topic was conducted using ^161^Tb, which primarily emits beta particles. In contrast, using ^99m^Tc allows us to study this phenomenon with a radionuclide that emits only Auger electrons and no other types of corpuscular radiation.

## 2. Results and Discussion

Current nuclear medicine practice primarily utilizes SSTR2-targeting radiopharmaceuticals that function as receptor agonists, such as ^99m^Tc-EDDA/HYNIC-TOC, ^177^Lu-DOTATATE, and ^68^Ga-DOTATOC. However, both preclinical and clinical development are increasingly concentrating on SSTR2 antagonists. [11,12]. This is due to the ability of antagonists to recognize more binding sites on the receptor [13]. This led to increased tumor uptake and better tumor-to-background contrast, resulting in enhanced image sensitivity and higher radiation doses to the tumor.

As mentioned in the introduction, a significant increase in cytotoxicity was observed for the SSTR2 receptor antagonist ^161^Tb-DOTA-LM3 compared to the agonist ^161^Tb-DOTATOC. This phenomenon can be attributed to the cytotoxic effects of Auger electrons on the cell membrane, where ^161^Tb-DOTA-LM3 accumulates. Since the radionuclide ^161^Tb emits both β^−^ particles and Auger electrons, these findings suggest that the cell membrane is a more sensitive target than the cytoplasm for the dense ionization produced by Auger electrons. Previous studies have shown that when targeting tumor clusters, ^161^Tb delivered 2- to 6-fold higher absorbed doses to cell membranes than ^177^Lu [14]. Up to now, no cytotoxicity studies have been conducted for SSTR2 antagonists labeled with Auger emitters that do not emit β^−^ particles. The commercial availability of a bioconjugate of the SSTR2 receptor agonist-TEKTROTYD and the antagonist—TECANT-1, ready for labeling with ^99m^Tc, allows for easy performing of these studies and evaluation of the possibility of therapeutic application of ^99m^Tc in the treatment of metastatic neuroendocrine tumors.

The structure of radiobioconjugates is presented in Figure 1. As shown in the figure, TEKTROTYD is a conjugate of octreotide molecule with HYNIC ligand, and TECANT-1 is a conjugate of LM3 (p-Cl-Phe-cyclo(DCys-Tyr-DAph(Cbm)-Lys-Thr-Cys)-D-Tyr-NH_2_) molecule with tetraamine linear ligand.

As expected, HYNIC and tetraamine were very effective ligands for ^99m^Tc. After reducing ^99m^TcO_4_ with SnCl_2_, TECANT-1 and TEKTROTYD were efficiently labeled with ^99m^Tc, achieving yields of 98.9 ± 0.4% and 94.4 ± 3.8%, respectively. After purification using a Sep-Pak C18 column, the radiochemical purity increased to 99.6 ± 0.4%. As reported in multiple studies, ^99m^Tc-TECANT-1 and ^99m^Tc-TEKTROTYD exhibited high stability in biological fluids, including human serum [15].

To compare the receptor binding of ^99m^Tc-TEKTROTYD and ^99m^Tc-TECANT-1, we calculated the specific binding of both radioconjugates by subtracting nonspecific binding from total binding. Figure 2 illustrates the relationship between specific binding and the concentration of the radioconjugates. In Appendix A Appendix A, we included experimental data on total binding and non-specific binding in the presence of a 2000-fold excess of octreotide, which allowed us to determine specific binding. In Appendix A, the K_D_ and B_max_ values are presented as determined from the saturation curves. Specific binding for the ^99m^Tc-labeled antagonist TECANT-1 is significantly greater than for the agonist TEKTROTYD. These results are consistent with previous preclinical studies, demonstrating much higher tumor accumulation of these non-internalizing SST analogues than for SSTR agonists. A possible explanation may be that antagonist radioligands may label a higher number of receptor-binding sites than agonist radioligands [16]. Scatchard analysis in SSTR2–transfected HEK293 cells showed more than 10 times the number of binding sites for the SST antagonist ^111^In-DOTA-BASS than for the SST agonist [^111^In-DTPA^0^,Tyr^3^,Thr^8^]-octreotide [11].

Due to the limited range of Auger electrons, internalization of radiobioconjugates is crucial for achieving optimal therapeutic effects. As shown in Figure 3, approximately 90% of ^99m^Tc-TEKTROTYD is internalized into AR42J cells through SSTR2 receptors while in the case of ^99m^Tc-TECANT-1 most of the radioconjugate is accumulated in the cell membrane. The situation is similar to the previously described analogous conjugates of ^161^Tb labeled octreotide (TOC) with DOTA, an agonist of the SSTR2 receptor, and the conjugate of DOTA with the LM3 peptide, an antagonist of this receptor. About 9% of the cellular uptake of radiolabeled DOTA-LM3 was internalized, whereas in the case of DOTATOC, the internalized fraction was much higher, about 81% [17]. The internalized fraction is increasing with time, from 1 h to 4 h. The differences in receptor internalization can be understood by recognizing that an agonist binds to a receptor, inducing internalization. This process involves the receptor being taken into the cell. In contrast, an antagonist binds to a cell membrane receptor but does not activate it, remaining attached to the cell membrane instead. The almost identical internalization of ^161^Tb-labeled DOTATOC and DOTA-LM3 conjugates and those of the peptides conjugated with ^99m^Tc-labeled HYNIC and N4 chelators shows that their properties do not depend on the type of attached chelator.

In the cytotoxicity studies of the short-range Auger electrons, considerable attention has been focused on delivering Auger electron emitting radionuclides to the cell nucleus, preferably close to DNA. This is the best way to achieve a therapeutic effect. Most publications on AE therapy have been devoted to this issue. However, further work has shown that internalization into cancer cells and delivery of Auger electron emitters to the cell nucleus is not necessary to kill the cancer cell, and the therapeutic effect can also be achieved indirectly through free radicals generated in the cytoplasm. It was found also that targeting the cell membrane has also been shown to be an effective strategy for killing cancer cells using AE [3,18,19].

To date, studies on somatostatin analogs as therapeutic radiopharmaceuticals have primarily focused on conjugates labeled with β^−^ emitters, such as ^177^Lu and ^90^Y [20] and ^161^Tb [21], which emits β^−^ radiation together with Auger electrons. The properties of agonist octreotide conjugates labeled with the radionuclide ^111^In have been studied in several works, primarily for imaging but also for Auger electron therapy. The ^111^In not only emits gamma quanta but also produces 7.4 Auger electrons. However, the therapeutic effects observed with these conjugates are less significant than those labeled with ^177^Lu or ^90^Y [22]. In a few works, Maecke et al. also studied ^111^In radioconjugates with SSTR receptor antagonists, but these studies were focused on tumor imaging and the therapeutic effect was not investigated [23]. In our studies, we decided to apply the differential levels of ^99m^Tc-TEKTROTYD and ^99m^Tc-TECANT-1 accumulation in the cell membrane and cytoplasm to evaluate which target is more effective for cell destruction. Figure 4 shows the results of cytotoxicity studies of ^99m^Tc-labeled SSTR2 receptor agonist and antagonist performed on AR42J cells. As shown in Figure 4, the toxicity of the ^99m^Tc-labeled receptor antagonist is two to three times higher than that of the agonist. The observed increase in cell viability after 72 h of incubation for agonist ^99m^Tc-TEKTROTYD seems illogical. However, we must take into account the short half-life of ^99m^Tc- 6 h and the long incubation time of 48 h and 72 h. ^99m^Tc has a cytotoxic effect only for the initial incubation period. In the case of ^99m^Tc-TEKTROTYD, which is located in the cytoplasm, emitted Auger electrons do not cause double-strand DNA breaks, but act indirectly through radicals generated by water radiolysis, which cause single-strand breaks in DNA. They are easily repairable and the cells could continue to divide. As a result, there was a significant increase in cell viability observed after 72 h. In the case of TECANT-1, no such effect was observed because the conjugate was bound to the receptor on the cell membrane and irreversibly damaged the membrane.

The obtained results clearly indicated that the cell membrane is a much better therapeutic target than the cytoplasm. This is confirmed by numerous results of studies comparing the cytotoxicity of DOTATATE (SSTR2 receptor agonist) radiobioconjugates and DOTA-LM3 (SSTR2 receptor agonist) labeled with ^177^Lu and ^161^Tb. Lu^3+^ and Tb^3+^ cations have almost identical chemical properties, but the radionuclide ^161^Tb, unlike ^177^Lu, is an emitter of β^−^ radiation and Auger electrons, while ^177^Lu basically emits only β^−^ radiation. It was observed that the non-internalizing conjugate ^161^Tb-DOTA-LM3 showed much more significant cytotoxicity than ^177^Lu-DOTA-LM3, while in the case of radiobioconjugates ^177^Lu-DOTATATE and ^161^Tb-DOTATATE internalizing to the cytoplasm, no significant differences were observed.

When comparing the cytotoxicity of the peptides octreotide and LM3 labeled with ^99m^Tc and ^161^Tb, we observe a significantly greater effect with the ^161^Tb radioconjugates. This difference is largely attributed to ^161^Tb being an emitter of low-energy β^−^ radiation, which has a much longer range than that of Auger and conversion electrons. Whether ^161^Tb accumulates in the cell membrane or cytoplasm, the emitted β^−^ particles can interact with DNA in a toxic manner. So, for the ^161^Tb-DOTA-LM3 radioconjugate, the observed toxicity is likely due to the cumulative effect of β^−^ particles damaging the DNA within the cell nucleus, along with the interaction of Auger electrons affecting the cell membrane. In contrast, for the ^99m^Tc-N4-LM3 radioconjugate (TECANT-1), we see only cytotoxicity effect associated with the destruction of the cell membrane.

Also, it is important to consider the conversion electrons (CE) emitted by ^161^Tb and ^99m^Tc as well. Both radionuclides emit CE with similar yield: 1.1 per decay for ^99m^Tc and 1.4 for ^161^Tb. These CE have a micrometer range, allowing them to interact with cell DNA. Therefore, the slight cytotoxicity associated with the agonist ^99m^Tc-TEKTROTYD may be attributed to its EC emitted by ^99m^Tc in the cytoplasm.

## 3. Materials and Methods

### 3.1. Chemical Reagents

The following chemical reagents—TEKTROTYD (20 micrograms kits) (HYNIC-(D-Phe^1^,Tyr^3^-Octreotide) and TECANT-1 (N_4_-p-Cl-Phe-cyclo(D-Cys-Tyr-DAph(Cbm)-Lys-Thr-Cys)-D-Tyr-NH_2_, where D-Aph(Cbm): D-4-amino-carbamoyl-phenylalanine)—were purchased from (Polatom, Otwock, Poland): Sodium hydroxide (NaOH, VWR CHEMICA, EC), Aceton (VWR CHEMICA, EC), Ethyl alcohol Absolut 99.8% (POCH, Gliwice, Poland), Methanol (VWR CHEMICA, EC), Ammonium acetate (VWR CHEMICALS, EC), ITLC-SG (chromatography paper, Serial/Lot No: 6446280-03, Agilent Technology, Santa Clara, CA, USA), N-(3-dimethylaminopropyl)-N-0-ethylcarbo-diimide hydrochloride (EDC, >99%), and Acetonitril, >99.9% for HPLC (Sigma-Aldrich, Saint-Quentin-Fallavier, France). The aqueous solutions were prepared using ultrapure deionized water 18.2 MW_cm (Hydrolab, Straszyn, Poland), Sep-Pak-18 classic cartridge (Waters Corporation, Milford, MA, USA), and Glycine (Sigma, Life Science, Buchs, Switzerland).

### 3.2. Cell Lines and Reagents for Biological Studies

AR-42-J (Epithelial-like cell isolated from pancreas of a rat with tumor) was purchased from American Type Culture Collection (ATCC, Rockville, MD, USA) and cultured according to the ATCC protocol (humidified atmosphere of 5% CO_2_ at 37 °C). Cells were cultured in F-12K (Kaighn’s Modification of Ham’s F-12 with L-glutamine) enriched with 10% heat-inactivated fetal bovine serum and antibiotics: penicillin and streptomycin (100 IU/mL). Trypsin EDTA solution C (0.25%) was used to detach the cells. All chemicals mentioned above were purchased from Capricorn Scientific GmbH (Ebsdorfergrund, Germany) and ATCC collection, USA. CellTiter96^®^ Aqueous One Solution Reagent (MTS compound) from Promega (Mannheim, Germany) and phosphate-buffered saline (PBS) from Capricorn Scientific GmbH (Ebsdorfergrund, Germany) were used for cell studies.

### 3.3. Radionuclide (Na^+^[^99m^Tc]TcO_4_^−^)

No-carrier-added Na^+99m^TcO_4_^−^ was obtained from a commercial ^99^Mo/^99m^Tc-generator (Polatom, Otwock, Poland). POLTECHNET—kit for elution was purchased from Polatom. The kit was composed of 16 vials with an eluent of 10 mL volume containing 9 mg/mL (0.9%) NaCl solution and 16 evacuated vials.

### 3.4. Instruments

The radioactivity of ^99m^Tc was counted by ATOMLAB™ 500, BIODEX, New York, NY, USA and Wizard2 Detector Gamma Counter (Wizard2 gamma counter, PerkinElmer, Waltham, MA, USA). The radiolabeling yield was determined by using TLC and a Storage Phosphor System Cyclone Plus (PerkinElmer, Waltham, MA, USA). The samples of in vitro studies, including affinity and internalization, were measured by Wizard2 Detector gamma counter. The MTS assay was performed by BioTek (SYNERGY H1, Winooski, VT, USA) and absorbance at 490 nm was measured for determination of cell metabolic activity percentage. The analysis was performed by BioTek Gen5 (Microplate Reader and Gen5 Imager Software, Winooski, VT, USA). Heater Grant (QBD2, Royston, Cambridgeshire, UK) was used in the incubation of the TEKTROTYD radiolabeling process.

### 3.5. Radiolabeling and Purification of TEKTROTYD

The radiolabeling process was performed according to the kit protocols, with minor modifications. The labeling kit consisted of two vials containing components necessary for the preparation of ^99m^Tc-TEKTROTYD. Vial I contains the active substance: HYNIC-[D-Phe^1^, Tyr^3^-Octreotide] trifluoroacetate, along with excipients including stannous chloride dihydrate, N-[Tris(hydroxymethyl)methyl]glycine (tricine), and mannitol. Vial II contains excipients such as ethylenediamine-N,N’-diacetic acid (EDDA), disodium phosphate dodecahydrate, sodium hydroxide, and either sodium hydroxide or hydrochloric acid for pH adjustment. A total of 4 mL of Na^+^[^99m^Tc]TcO_4_^−^ was eluted from the ^99^Mo/^99m^Tc generator using a 0.9% NaCl solution, which was then placed in an evacuated vial. The eluted activity was measured using a dose calibrator. The solution was then added to vial II and gently mixed for 15 s. Subsequently, the contents of vial II were transferred to vial I and mixed for an additional 30 s. The sample was then heated at 100 °C for 10 min to facilitate ethanol evaporation, followed by cooling to room temperature.

For purification, the obtained sample was passed through a Sep-Pak C18 column, and the column was washed with 1 mL of water. Finally, ^99m^Tc-TEKTROTYD was eluted with 1 mL of 90% EtOH and collected in separate glass vials.

For quality control, the radiopeptide solution was heated to evaporate EtOH. After evaporation, the radiocompound was dissolved in 500 µL of 0.01M PBS.

### 3.6. Radiolabeling of TECANT-1

The radiolabeling process was performed according to the modified kit protocol. Briefly, the Na^+^[^99m^Tc]TcO_4_^−^ was eluted from the ^99^Mo/^99m^Tc generator (Polatom, Świerk, Poland) using a 0.9% NaCl solution and added to a mixture of 25 μL of 0.5 M phosphate buffer, and 5 μL of 0.1M trisodium citrate. This was followed by the addition of TECANT-1 (20 μg, in H_2_O) and 5 μL SnCl_2_ in EtOH. The radiolabeling solution was incubated at RT for 10 min. After this time, the vial was opened, and 10 µL of the solution was taken to determine the pH of the radiopeptide solution. The sample was then heated at 100 °C for 10 min to facilitate ethanol evaporation, and cooled to room temperature. The purification of ^99m^Tc-TECANT-1 was performed using the same procedure as for the ^99m^Tc-TEKTROTYD.

### 3.7. Binding Affinity Assay

The SSTR2 binding affinity of ^99m^Tc-TEKTROTYD (^99m^Tc-EDDA/HYNIC-3Tyr-octreotide and ^99m^Tc-TECANT-1 (^99m^Tc-N_4_-p-Cl-Phe-cyclo(D-Cys-Tyr-D-4-amino-Phe(carbamoyl)-Lys-Thr-Cys)D-Tyr- NH_2_) was determined by saturation binding affinity assay. AR-42-J cells were seeded in 6-well plates (7 × 10^5^ cells/well for octreotide) and 12-well plates (4 × 10^5^ cells/well for LM-3) and incubated at 37 °C for 48 h. Following incubation, the cells were washed twice with PBS and incubated for 2 h with increasing concentrations of radiopeptides (0.004–20 nM). To assess non-specific binding, wells were incubated with an additional 2000-fold excess of unlabeled octreotide acetate. Following incubation, the cells were washed with PBS, collected into tubes and then lysed twice using 800 μL of 1 M NaOH (for 12-well plate) and 1.5 mL of 1 M NaOH (for 6-well plates). The lysates were collected in separate tubes and radioactivity was measured using Wizard2 gamma counter, PerkinElmer, Waltham, MA, USA. The lysates were transferred to separate tubes and radioactivity was quantified using Wizard2 gamma counter.

### 3.8. Internalization Assay

The intracellular retention of ^99m^Tc-TEKTROTYD and ^99m^Tc-TECANT-1 in SKOV-3 cells was assessed at various time intervals. Briefly, AR42J cells (7 × 10^5^ cells/well) were seeded in 6-well plates and incubated for 2 days. Following this incubation, the cells were washed twice with cold PBS and then treated with 5 nM of tested compounds. To avoid internalization, the cells were incubated at 4 °C for 1 h. Non-specific binding was assessed by adding a 1000-fold molar excess of unlabeled octreotide acetate. After incubation, unbound fraction was collected in tubes, and the cells were washed twice with 1 mL of cold PBS. Fresh medium was then added, and the cells were incubated at 37 °C for up to 24 h. At each time interval, the second medium portion was collected, and cells were washed twice with 1 mL of PBS. Next, the membrane-bound fraction was collected using 0.05 M glycine–HCl buffer (pH 2.8). Finally, the internalized fraction was obtained by lysing the cells with 1.5 mL of 1 M NaOH.

### 3.9. Cytotoxicity Studies

A cytotoxicity assay was performed using the AR42J cell line. A total of 10^4^ cells were seeded in 96-well plate and incubated overnight at 37 °C in a 5% CO_2_ atmosphere. The following day, the medium was removed and the cells were treated with radiocompounds at concentrations of 100, 50, 25, 12.5 MBq/mL The cells were then incubated for 24, 48, and 72 h. At each specified time point, 20 µL of MTS reagent was added to cells and incubated for 2 h. Absorbance at 490 nm was measured to estimate the percentage of cell metabolic activity. At each specified time point, 20 µL of MTS reagent was added to the cells and incubated for 2 h, after which absorbance at 490 nm was measured. To assess the percentage of metabolically active cells, control experiments were conducted using untreated cells as a negative control. The average absorbance of these control wells was subtracted from the absorbance of wells containing treated cells.

### 3.10. Statistical Analysis

GraphPad Prism version 8.4.3 software (GraphPad Software Inc., San Diego, CA, USA) was used for the statistical analysis of the experimental data. Data points and SD are from at least two or more measurements.

## 4. Conclusions

The therapeutic application of ^99m^Tc offers several potentially advantageous features, including an optimal half-life, stable daughter nuclides, and Auger electron energies that are suitable for effective radiotherapy of targeted tumors. Additionally, ^99m^Tc is readily available and inexpensive. However, its limited efficiency in the emission of Auger electrons per decay—when compared to isotopes like ^111^In and ^125^I—may restrict its use in targeted Auger electron therapy.

Based on the cellular studies presented in this work, it can be concluded that the SSTR2 receptor antagonist ^99m^Tc-TECANT-1 demonstrates a significantly enhanced cytotoxic effect compared to the agonist ^99m^Tc-TEKTROTYD. This difference in cytotoxicity is even more pronounced than when using the same peptides labeled with ^161^Tb, where an additional effect from beta particle emission also contributes.

It is important to note that ^99m^Tc does not emit beta radiation or high-energy gamma quanta, which allows for very high radiation doses to be administered without causing significant side effects. Further dosimetry calculations and in vivo studies are necessary to determine the potential effectiveness of Auger electrons emitted from ^99m^Tc-TECANT-1 in targeted radiotherapy for neuroendocrine tumors.

## Figures and Tables

**Figure 1 molecules-30-01715-f001:**
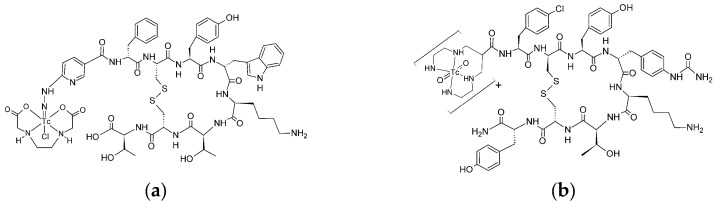
Structure of ^99m^Tc labeled TEKTROTYD (**a**) and TECANT-1 (**b**).

**Figure 2 molecules-30-01715-f002:**
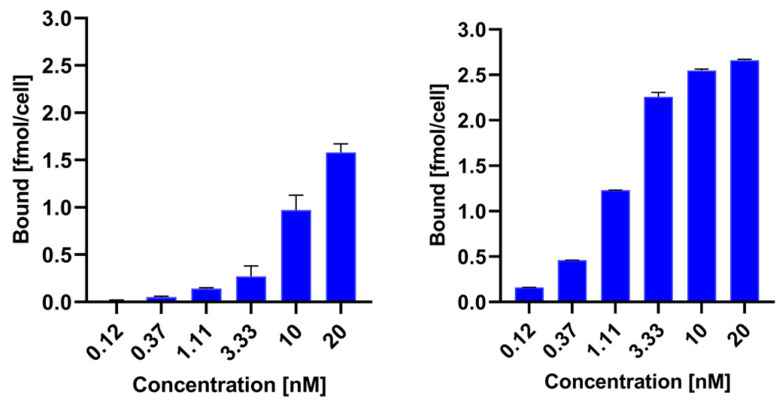
Specific binding of ^99m^Tc-TEKTROTYD and ^99m^Tc-TECANT-1 on AR42J cells.

**Figure 3 molecules-30-01715-f003:**
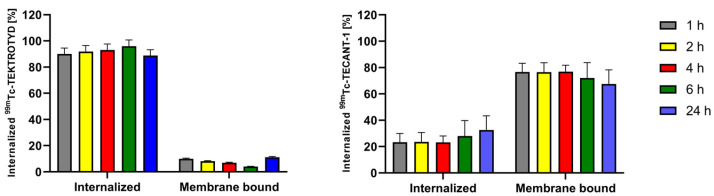
Internalization of ^99m^Tc-labeled peptide agonist (**left graph**) and antagonist (**right graph**) of the sstr2 receptor on AR42J cells.

**Figure 4 molecules-30-01715-f004:**
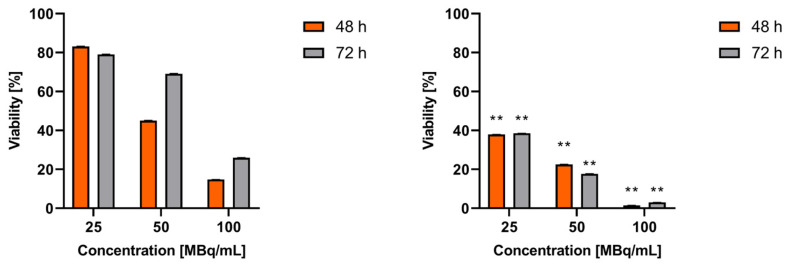
Cytotoxicity of ^99m^Tc-labeled SSTR2 receptor agonist TEKTROTYD (**left graph**) and antagonist TECANT-1 (**right graph**) for AR42J cells. The data results are expressed in mean ± standard deviation. One-way ANOVA analysis was used for multiple group comparisons, followed by Tukey’s test. The cell experiments were repeated at least 2 times. ** indicates that the difference is statistically significant compared to the control group (untreated cells—100% of viability) with a *p* < 0.01.

## Data Availability

The datasets presented during the current study are available from the corresponding authors on reasonable request.

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
