# Peer review of "Cytotoxicity Comparison of ^99m^Tc-Labeled Peptide Antagonist and Agonist Targeting the SSTR2 Receptor in AR42J Cells"

_molecules, 2025, doi:10.3390/molecules30081715_

Round 1

Reviewer 1 Report

Comments and Suggestions for Authors

Aleksander Bilewicz et al. describe the evaluation of the radiobiological effects induced by a SST2 agonist and SST2 antagonist labeled with 99mTc, aiming to take advantage from the localized effects associated with the short-range Auger electrons (AEs) emitted by 99mTc. The work was inspired by the results previously reported by Müller et al. for the same agonist and antagonist peptides labeled with 161Tb, which have shown that the antagonist led to more pronounced radiobiological effects due most probably to the irradiation of the cell membrane by the emitted AEs.

It is an important topic of research deserving further studies, to have a better understanding of the promising results obtained for radiolabeled antagonists in AE radiopharmaceutical therapy of cancer. However, the reported work has several limitations, namely due to missing control experiments and to the need of performing some additional experiments. Before the manuscript can be considered in appropriate form for publications, the authors must address the following issues:

  • Page 1, Abstract, line 23: I suggest to remove “highly-radioactive”, as it will depend on the used radioactivity and is not relevant in this context.

  • Page 2, lines 52 and 33: The authors claim that “it is possible to inject approximately tenfold the radioactivity of Auger emitters compared to β- particle emitters without toxic side, particularly affecting the bone marrow.” This points needs to be discussed more carefully, as many AE emitters also emit gamma photons that restrict the injectable dose namely due to radiation protection issues during patient management. In other parts of the manuscript where this point is discussed, these limitations need to be taken into consideration.

  • Page 3, Line 94: Provide a reference for the reported clinical study.

  • Page 3, Line 97: The authors claim that 161Tb emit 2.27 AEs per decay, which is not true. Please, check the right value that is mentioned in the following references:

  1. Bolcaen J, Gizawy MA, Terry SYA, Paulo A, Cornelissen B, Korde A, et al. Marshalling the Potential of Auger Electron Radiopharmaceutical Therapy. J Nucl Med. 2023;64:1344–51;
  2. Van Laere C, Koole M, Deroose CM, De Voorde MV, Baete K, Cocolios TE, et al. Terbium radionuclides for theranostic applications in nuclear medicine: from atom to bedside. Theranostics. 2024;14:1720–43.

These references should be included and the number of emitted AEs corrected in accordance.

  • Page 4, Figure 2: The legend of Figure 2 some more information. Why the authors dis not determine the binding affinity with the obtained data? The data of the non-specific binding should also be presented, at least in the SI.

  • Page 5, Lines 174-175: The labeling radionuclide used in the cellular studies with the DOTA derivatives must be mentioned.

  • Page 6, Figure 4/Cell viability studies:

i) The viability values are relative to control experiments? Control experiments were performed? This needs to be mentioned and clarified.

ii) The data for the 24 h incubation with the radiocompounds are not presented, although the experiments were run. Why?

iii) For the agonist the shorter incubation time of 48 h has led to reduced viability compared to the incubation time of 72 h. By contrast, the antagonist led to almost equal viability values for both incubation times. These results need to be further discussed.

iv) The statistical significance for the differences of the cell viability data (between the two radiocompounds and incubation times) must be calculated and presented.

v) I encourage the authors to perform also clonogenic cell survival assays to have a better insight on the differences of the radiobiological effects caused by the 99mTc-labeled agonist and antagonist.

Author Response

It is an important topic of research deserving further studies, to have a better understanding of the promising results obtained for radiolabeled antagonists in AE radiopharmaceutical therapy of cancer. However, the reported work has several limitations, namely due to missing control experiments and to the need of performing some additional experiments. Before the manuscript can be considered in appropriate form for publications, the authors must address the following issues:

We very grateful for the reviewer's comments. These were useful for improving the manuscript.

We agree with the reviewer that the work should have additional experiments, e.g. double-strand break studies, confocal microscopy, cell cycle, etc. Unfortunately, we had a very limited amount of TECANT-1 conjugate and did not have enough to perform these experiments. TECANT-1 production in Polatom has already ended and the producer does not have any more of the product. Despite these limitations, we believe that the results we have obtained will still be of interest to readers, particularly those focused on Auger electron therapy.

  • Page 1, Abstract, line 23: I suggest to remove “highly-radioactive”, as it will depend on the used radioactivity and is not relevant in this context.

We changed “highly-radioactive” for “therapeutic doses”, thank you

  • Page 2, lines 52 and 33: The authors claim that “it is possible to inject approximately tenfold the radioactivity of Auger emitters compared to β- particle emitters without toxic side, particularly affecting the bone marrow.” This points needs to be discussed more carefully, as many AE emitters also emit gamma photons that restrict the injectable dose namely due to radiation protection issues during patient management. In other parts of the manuscript where this point is discussed, these limitations need to be taken into consideration.

This sentence concerns the comparison with beta emitters. We agree with the reviewer that gamma emission must be taken into account. High activities can certainly be used with Auger emitters with a high e/photon ratio emission (Bernhardt P, et al., Acta Oncol. 2001, 40, 602–8.). We have added the appropriate sentence to the text.

  • Page 3, Line 94: Provide a reference for the reported clinical study.

 Done

  • Page 3, Line 97: The authors claim that 161Tb emit 2.27 AEs per decay, which is not true. Please, check the right value that is mentioned in the following references:

  1. Bolcaen J, Gizawy MA, Terry SYA, Paulo A, Cornelissen B, Korde A, et al. Marshalling the Potential of Auger Electron Radiopharmaceutical Therapy. J Nucl Med. 2023;64:1344–51;
  2. Van Laere C, Koole M, Deroose CM, De Voorde MV, Baete K, Cocolios TE, et al. Terbium radionuclides for theranostic applications in nuclear medicine: from atom to bedside. Theranostics. 2024;14:1720–43.

These references should be included and the number of emitted AEs corrected in accordance.

Various data are presented in the literature on Auger electron emission by 161Tb. In the previous version, I included the data with the number of AEs 2.27 for AE energy > 3 keV. However, at the reviewer's suggestion, I included the data for all AEs reported in the last two review articles (10.9). Appropriate references were added.

  • Page 4, Figure 2: The legend of Figure 2 some more information. Why the authors dis not determine the binding affinity with the obtained data? The data of the non-specific binding should also be presented, at least in the SI.

Our results related to total binding and nonspecific binding are presented in the supplementary materials. (Figure S1).

  • Page 5, Lines 174-175: The labeling radionuclide used in the cellular studies with the DOTA derivatives must be mentioned.

            Done

  • Page 6, Figure 4/Cell viability studies:

  1. i) The viability values are relative to control experiments? Control experiments were performed? This needs to be mentioned and clarified.

As a control, untreated cells were used, and their viability was determined as 100%. This information has been added to the text in section 3.9 Cytotoxicity Studies. 

  1. ii) The data for the 24 h incubation with the radiocompounds are not presented, although the experiments were run. Why?

The 24-hour incubation results were not obtained due to an incubator failure. Unfortunately, we could not repeat them because there was a limited amount of TECANT-1 available. Therefore, we have included only the results from the 48-hour and 72-hour incubations.

iii) For the agonist the shorter incubation time of 48 h has led to reduced viability compared to the incubation time of 72 h. By contrast, the antagonist led to almost equal viability values for both incubation times. These results need to be further discussed.

The increase in cell viability after 72 hours of incubation for agonist 99mTc-TEKTROTYD seems illogical. However, we must take into account the short half-life of 99mTc - 6 h and the long incubation time of 48 h and 72 h. 99mTc-TEKTROTYD has a cytotoxic effect only for the initial incubation period. 99mTc-TEKTROTYD, which is located in the cytoplasm, emitted Auger electrons do not directly cause double-strand DNA breaks; instead, they indirectly lead to single-strand breaks in DNA through radicals generated by water radiolysis. They are easily repairable, and the cells, after decay of 99mTc, could continue to divide.

In the case of TECANT-1, there was no observed such effect because the conjugate bound to the receptor on the cell membrane and irreversibly damaged it.

We have added the appropriate discussion to the manuscript

  1. iv) The statistical significance for the differences of the cell viability data (between the two radiocompounds and incubation times) must be calculated and presented.

Thank you. The statistical significance was calculated (Figure 4).

  1. v) I encourage the authors to perform also clonogenic cell survival assays to have a better insight on the differences of the radiobiological effects caused by the 99mTc-labeled agonist and antagonist.

Thank you for your suggestion. We fully agree that clonogenic cell survival assays would provide valuable additional insights into the radiobiological effects of the 99mTc-labeled agonist and antagonist. In fact, we routinely perform such experiments in our laboratory. However, due to the limited availability of TECANT-1, we were unable to conduct additional studies. This compound is currently unavailable, which restricted our ability to perform further assays. Given these constraints, we selected the MTS assay as the most optimal method for evaluating cell viability in this study.

Reviewer 2 Report

Comments and Suggestions for Authors

The manuscript by Shanjani et al. describes a comparison of the cytotoxicity between the two 99mTc-labelled somatostatin agonist and antagonist on AR4-2J cells. The manuscript contains several shortcomings and cannot be accepted in the current form.

As general comment, the manuscript contains repetitive concepts in introduction and discussion, lacks a proper discussion and does not present convincing experimental data to support the hypothesis of the authors. Importantly, important references are missing.

In the introduction the authors should reduce redundant references to 161Tb-DOTA-LM3 as a comparative ligand and clearly state the aim of their study (which is only mentioned in the results and discussion sections). They should also include published data on other 99mTc-labeled somatostatin ligands, including TECANT1, as one example. The novelty of the manuscript, which appears to rely on the therapeutic potential of the auger electron emitted by 99mTc, should be discussed in comparison with other well-known auger emitters, such as 111In, which is also used for labeling somatostatin ligands, agonists and antagonists.

The results section does not report all the results. For example, saturation binding studies are used to calculate KD and Bmax, but these results are not provided.

The experimental methods are poorly described and not well executed.

Saturation binding assay. The authors should specify the temperature of incubation, explain the reason for using different well plates, and clarify why the same concentration range was used for both radioligands. Moreover, the authors should explain why they incubated the radioligands for 2 hours in the binding assay and 1 hour in the internalization assay. The incubation time for the saturation binding study should be determined based on internalization studies conducted at different time points to identify the appropriate timing when ‘steady-state’ equilibrium is reached.

The design of the internalization experiments is unclear. Can the authors explain the reason for incubating at 4°C to avoid internalization during the internalization study? 

Did the authors check the number of AR4-2J cells after incubation at 4°C? This cell line is highly sensitive to temperature.

In the cytotoxicity studies, the authors should clarify how the percentage of cell viability was calculated. Did they use untreated cells as a control?

Comments on the Quality of English Language

This manuscript presents many redundant sentences and repetitive concepts

Author Response

The manuscript by Shanjani et al. describes a comparison of the cytotoxicity between the two 99mTc-labelled somatostatin agonist and antagonist on AR4-2J cells. The manuscript contains several shortcomings and cannot be accepted in the current form.

As general comment, the manuscript contains repetitive concepts in introduction and discussion, lacks a proper discussion and does not present convincing experimental data to support the hypothesis of the authors. Importantly, important references are missing.

Following the reviewer suggestion, we have removed repetitive paragraphs in the Introduction and Results and discussion. We also added some literature references. We agree with the reviewer that the does not present convincing experimental data to support the hypothesis. It would be beneficial if the work had additional experiments, e.g. double-strand break studies, confocal microscopy, cell cycle, etc. Unfortunately, we had a very limited amount of TECANT-1 conjugate and did not have enough to perform these experiments. TECANT production in Polatom has already ended and the producer does not have any more of the product. Despite these limitations, we believe that the results we have obtained will still be of interest to readers, particularly those focused on Auger electron therapy.

In the introduction, the authors should reduce redundant references to 161Tb-DOTA-LM3 as a comparative ligand and clearly state the aim of their study (which is only mentioned in the results and discussion sections). They should also include published data on other 99mTc-labeled somatostatin ligands, including TECANT1, as one example. The novelty of the manuscript, which appears to rely on the therapeutic potential of the auger electron emitted by 99mTc, should be discussed in comparison with other well-known auger emitters, such as 111In, which is also used for labeling somatostatin ligands, agonists and antagonists.

In response to the reviewer's suggestion, in the introduction, we limited references to the 161Tb-DOTA-LM3. We have moved the objectives of our studies from the results section to the introduction. We have also added a whole paragraph regarding 111In-labeled antagonists and agonists of sstr receptors.

The results section does not report all the results. For example, saturation binding studies are used to calculate KD and Bmax, but these results are not provided.

We added the graphs of saturation binding experiments and also determined Kd and Bmax values. The results are presented in Supplementary materials.

The experimental methods are poorly described and not well executed.

Thank you. We improved the experimental methods part.

Saturation binding assay. The authors should specify the temperature of incubation, explain the reason for using different well plates, and clarify why the same concentration range was used for both radioligands. Moreover, the authors should explain why they incubated the radioligands for 2 hours in the binding assay and 1 hour in the internalization assay. The incubation time for the saturation binding study should be determined based on internalization studies conducted at different time points to identify the appropriate timing when ‘steady-state’ equilibrium is reached.

We have clarified the incubation temperature and added this information to the text in paragraph 3.7, Binding Affinity Assay. Different well plates were used due to the limited quantity of TECANT-1 available. Since this compound is not currently accessible, we aimed to minimize the amount required for the experiment, which is why we chose to use 12-well plates instead of the usual 6-well plates typically used by us for such experiments.

The binding and internalization assays are two distinct experiments. In the internalization assay, we were focused on examining the kinetics of internalization at different time points to assess the process over time. In contrast, the binding affinity study is primarily concerned with the interaction between the radioligands and the receptors, and the exact incubation time does not significantly impact the results. For the binding assay, we did not consider whether the compound was on the membrane or inside the cell, as that was not relevant for the purpose of the experiment. However, we found (based on our experiments with the use of various cancer cells) that 1 hour of incubation is sufficient to reach reliable results for the binding affinity assay.

The design of the internalization experiments is unclear. Can the authors explain the reason for incubating at 4°C to avoid internalization during the internalization study? 

Thank you for your question. The cells were incubated at 4°C during the internalization experiments to prevent internalization while still allowing binding of the compounds to the cell membrane. At this temperature, endocytosis and other internalization processes are inhibited, ensuring that any compounds binding to the cell remain on the membrane. At 37°C, the internalization process begins. This approach allows us to confidently determine the percentage of internalized compounds.

Did the authors check the number of AR4-2J cells after incubation at 4°C? This cell line is highly sensitive to temperature.

We did not specifically check the number of AR4-2J cells after incubation at 4°C. However, we have worked with various cell lines, including AR4-2J cells, and have never observed any issues related to temperature sensitivity during short incubations at 4°C. Given the brief incubation time, we do not expect significant changes in cell morphology. In our previous experiments with AR4-2J cells, the cells remained viable and exhibited no noticeable changes.

In the cytotoxicity studies, the authors should clarify how the percentage of cell viability was calculated. Did they use untreated cells as a control?

As a control, untreated cells were used, and their viability was determined as 100%. This information has been added to the text in section 3.9 Cytotoxicity Studies. 

Round 2

Reviewer 1 Report

Comments and Suggestions for Authors

The authors addressed most of the raised questions and justified the difficulties to perform further experiments.

For these reasons, I am happy to recommend the publication of the manuscript in its current form.

Author Response

The reviewer accepted all our responses in the first round.

Reviewer 2 Report

Comments and Suggestions for Authors

I would like to thank the authors for their efforts in considering the previous suggestions and for improving the introduction. However, I believe there may be a critical misunderstanding regarding the experiments conducted, specifically the internalization and binding assays. These assays are central to the study, and, in my opinion, this misunderstanding prevents the manuscript from being accepted.

While I fully agree with the authors that internalization and binding assays are distinct experiments and I appreciate their general explanation of both, I have concerns regarding their execution and the way the results have been reported.

Regarding the internalization assay, it is performed to measure internalization and not to prevent it. If the internalization process is inhibited (at 4°C), it is not confidently possible to determine the percentage of internalized compounds. Moreover, generalla, the results should be expressed as % of internalized activity over the added activity. In this context, Figure 2 presents the percentage of radioligand internalized, but it is unclear what exactly this percentage represents.

For the binding affinity study, it is essential to conduct the experiment at 4°C to avoid internalization. At 37°C, internalization would likely occur, particularly with agonists, which would lead them to be internalized rather than remaining at the cell membrane. This would make it difficult to accurately measure receptor-ligand interactions. Additionally, the incubation time must be carefully controlled to ensure equilibrium between ligand bind and dissociate fraction.

The supplemental information clearly indicates  that internalization of the agonists predominates in the binding affinity assay, complicating the calculation of Bmax and KD values. 

Author Response

I would like to thank the authors for their efforts in considering the previous suggestions and for improving the introduction. However, I believe there may be a critical misunderstanding regarding the experiments conducted, specifically the internalization and binding assays. These assays are central to the study, and, in my opinion, this misunderstanding prevents the manuscript from being accepted.

While I fully agree with the authors that internalization and binding assays are distinct experiments and I appreciate their general explanation of both, I have concerns regarding their execution and the way the results have been reported.

Regarding the internalization assay, it is performed to measure internalization and not to prevent it. If the internalization process is inhibited (at 4°C), it is not confidently possible to determine the percentage of internalized compounds. Moreover, generalla, the results should be expressed as % of internalized activity over the added activity. In this context, Figure 2 presents the percentage of radioligand internalized, but it is unclear what exactly this percentage represents.

For the binding affinity study, it is essential to conduct the experiment at 4°C to avoid internalization. At 37°C, internalization would likely occur, particularly with agonists, which would lead them to be internalized rather than remaining at the cell membrane. This would make it difficult to accurately measure receptor-ligand interactions. Additionally, the incubation time must be carefully controlled to ensure equilibrium between ligand bind and dissociate fraction.

The supplemental information clearly indicates  that internalization of the agonists predominates in the binding affinity assay, complicating the calculation of Bmax and KD values. 

You are right that the procedure for internalization at 4°C presented in our publication seems illogical at first glance. However, it is often used. In our experiment, we first add the radiolabeled compound to the cells and incubate them at 4°C, where only the binding process occurs. This step ensures that any detected activity at this stage represents surface-bound ligand. We then remove the medium, replace it with fresh medium, and incubate the cells at 37°C for various time points to initiate and measure internalization.

It is important to clarify that internalization refers to the fraction of ligand that was initially bound to the membrane and subsequently partially internalized, rather than the total amount of ligand added to the cells. Therefore, the internalized fraction should be expressed relative to the membrane-bound activity rather than the total added activity. To summarize, it means that activity bound to membrane at 4oC is equals 100%.

The same procedure is also used in other publications like: Sinnes et al. 68Ga, 44Sc and 177Lu-labeled AAZTA5 -PSMA617: synthesis, radiolabeling, stability and cell binding compared to DOTA-PSMA-617 analogues. EJNMMI Radiopharmacy and Chemistry (2020) 5:28. Also in the IAEA GUIDANCE FOR PRECLINICAL STUDIES WITH RADIOPHARMACEUTICALS (IAEA RADIOISOTOPES AND RADIOPHARMACEUTICALS SERIES No. 8 p. 98 (www.iaea.org/publications) mentioned:

“Binding reactions are typically carried out at physiological temperature (37°C). To inhibit kinetics of cellular internalization/turnover of the tracer and specifically assess binding to membrane receptors, the reaction set may be incubated at 4–8°C. In such a case, at the end of the incubation the cells should be carefully examined under a microscope to assess possible damage from exposure to the lower temperature”.

However, based on the literature, most binding studies are conducted at 37°C (sometimes at 25oC or 30oC), where membrane-bound and internalized fractions are not differentiated. Below are some relevant examples:

  1. Mark Soave et al. (2016). Use of a new proximity assay (NanoBRET) to investigate the ligand-binding characteristics of three fluorescent ligands to the human β1-adrenoceptor expressed in HEK-293 cells. Pharmacol Res Perspect, 4(5), e00250. DOI: 10.1002/prp2.250.
  2. B. Bei Yao et al. (2006). Use of an inverse agonist radioligand [3H]A-317920 reveals distinct pharmacological profiles of the rat histamine H3 receptor. Neuropharmacology, 50(4), 463-472. DOI: 10.1016/j.neuropharm.2005.10.008.
  3. E. Martinez-Pinilla et al. (2017). Binding and Signaling Studies Disclose a Potential Allosteric Site for Cannabidiol in Cannabinoid CB2 Receptors. Front. Pharmacol., 8, 744. DOI: 10.3389/fphar.2017.00744.

Therefore, of these two procedures, we selected for our study the procedure where the binding is first performed at 4°C and then the internalization process is carried out at 37°C.

Round 3

Reviewer 2 Report

Comments and Suggestions for Authors

The experiment described by the authors is a redistribution of the cell membrane-bound radioligand after allowing cell binding for 2 hours at 4°C. In this case, the authors should report, alongside the internalized and membrane fractions, the dissociated fraction. The total should add up to 100%, considering that after the addition of fresh medium and incubation at 37°C, the membrane-bound radioligand will redistribute among the three fractions: internalized, membrane-bound, and dissociated. So, the graphic should report the 'Radioligand distrubution as % of the control (bound fraction at 4°C for 2h). 

I am confident that the authors have verified the reliability of their raw data, especially considering that the percentage of the bound fraction for a somatostatin-based radioagonist at 4°C is nearly negligible.